# Maternal–Neonatal Outcomes of Obstetric Deliveries Performed in Negative Pressure Isolation Rooms during the COVID-19 Omicron Variant Pandemic in Taiwan: A Retrospective Cohort Study of a Single Institution

**DOI:** 10.3390/jcm11185441

**Published:** 2022-09-16

**Authors:** Yi-Chiao Liao, Ping-Chung Wu, Li-Chun Chiu, Ho-Yen Chueh, Yu-Ning Chen, Yen-Chang Lee, Wen-Fang Li, Chi-Yuan Chiang, Chin-Chieh Hsu, Hsiu-Huei Peng, An-Shine Chao, Shuenn-Dyh Chang, Po-Jen Cheng, Meng-Chen Hsieh, Yao-Lung Chang

**Affiliations:** 1Department of Obstetrics and Gynecology, Chang Gung Memorial Hospital, Chang Gung University College of Medicine, Taoyuan 333, Taiwan; 2Division of Neonatology, Department of Pediatrics, Chang Gung Memorial Hospital, Chang Gung University College of Medicine, Taoyuan 333, Taiwan

**Keywords:** COVID-19, omicron variant, delivery, maternal outcomes, perinatal outcomes, respiratory distress

## Abstract

**Objective:** To investigate the maternal–neonatal outcomes of obstetric deliveries performed in negative pressure isolated delivery rooms (NPIDRs) during the coronavirus disease 2019 (COVID-19) omicron variant pandemic period in a single tertiary center in northern Taiwan. **Methods:** Confirmed positive and suspected-positive COVID-19 cases delivered in NPIDRs and COVID-19-negative mothers delivered in conventional delivery rooms (CDRs) in the period of 1 May 2022 to 31 May 2022 during the COVID-19 omicron variant pandemic stage were reviewed. The maternal–neonatal outcomes between the two groups of mothers were analyzed. All deliveries were performed following the obstetric and neonatologic protocols conforming to the epidemic prevention regulations promulgated by the Taiwan Centers for Disease Control (T-CDC). Multiple gestations, deliveries at gestational age below 34 weeks, and major fetal anomalies were excluded from this study. **Results:** A total of 213 obstetric deliveries were included. Forty-five deliveries were performed in NPIDRs due to a positive COVID-19 polymerase chain reaction (PCR) test (*n* = 41) or suspected COVID-19 positive status (*n* = 4). One hundred and sixty-eight deliveries with negative COVID-19 PCR tests were performed in CDRs. There was no statistical difference in maternal characteristics between the two groups of pregnant women. All COVID-19-confirmed cases either presented with mild upper-airway symptoms (78%) or were asymptomatic (22%); none of these cases developed severe acute respiratory syndrome. The total rate of cesarean section was not statistically different between obstetric deliveries in NPIDRs and in CDRs (38.1% vs. 40.0%, *p* = 0.82, respectively). Regardless of delivery modes, poorer short-term perinatal outcomes were observed in obstetric deliveries in NPIDRs: there were significant higher rates of neonatal respiratory distress (37.8% vs. 10.7%, *p* < 0.001, respectively), meconium-stained amniotic fluid (22.2% vs. 4.2%, *p* < 0.001, respectively) and newborn intensive care unit admission (55.6% vs. 8.3%, *p* < 0.001, respectively) in obstetric deliveries performed in NPIDRs than in CDRs. Maternal surgical outcomes were not significantly different between the two groups of patients. There was no vertical transmission or nosocomial infection observed in COVID-19 confirmed cases in this study period. **Conclusions:** Our study demonstrates that obstetric deliveries for positive and suspected COVID-19 omicron-variant cases performed in NPIDRs are associated with poorer short-term perinatal outcomes. Reasonable use of personal protective equipment in NPIDRs could effectively prevent nosocomial infection during obstetric deliveries for pregnant women infected with the COVID-19 omicron variant.

## 1. Introduction

Coronavirus disease 2019 (COVID-19), which is caused by a newly identified coronavirus, was first isolated in Wuhan, China, in December 2019. The global outbreak of COVID-19 has been unprecedented, with more than 550 million cumulative confirmed cases worldwide. Over 6 million lives were ravished by the 2019 novel coronavirus (2019-nCoV) [1]. Taiwan has experienced two waves of COVID-19 endemic. Announced by the Central Epidemic Command Center (CECC), the first wave was when Taiwan entered the community transmission stage of COVID-19 on 16 May 2021; the second wave of COVID-19 transmission in Taiwan was with the omicron variant, peaking in May 2022, with 40 to 100 thousand newly confirmed cases per day. So far, until 6 August 2022, a total of 4,680,681 laboratory-confirmed COVID-19 cases has been reported by Taiwan’s CEEC.

Since January 2020, the beginning of the global COVID-19 outbreak, the obstetric department of Chang Gung Memorial Hospital, Linkou Branch (Linkou CGMH), has established negative pressure isolated delivery rooms (NPIDRs) [2,3]. The NPIDRs were built in January 2020, costing approximately TWD 5,300,000, and are composed of four delivery rooms with anterooms and one recovery room; they are equipped with independent air conditioners, negative pressure facilities, and all other equipment that a conventional delivery room (CDR) requires. In response to the increasing levels of community transmission in Taiwan, a government-funded three-day validity COVID-19 polymerase chain reaction (PCR) test before admission was proposed in May 2021 to facilitate obstetric care; the PCR test validity was changed to 48 h in April 2022 due to the second wave of the COVID-19 outbreak in Taiwan. Our institution followed the strict guidelines decreed by the T-CDC to perform patient partitions, establish a cross-team management protocol, and provide healthcare services for confirmed or suspected COVID-19-positive cases with the rational use of personal protective equipment (PPE) in negative pressure rooms. Healthcare personnel (HCP) are under stress related to the time-consuming process of putting on PPE, possibly limited operative visual fields under PPE, and the risk of in-hospital spread of the COVID-19 infection.

Previous studies outside Taiwan have shown that COVID-19 infection in pregnancy is associated with adverse maternal and neonatal outcomes due to maternal physiologic changes and alterations of immune function during pregnancy [4,5,6,7]. However, information regarding clinical characteristics, maternal–fetal outcomes, and the potential vertical transmission rate of COVID-19 in pregnancy, especially of the omicron variant, among the Taiwanese population is limited, and the evidence is uncertain [3,8,9,10,11]. Limited practical experience for the obstetric management of COVID-19-infected pregnant women who have the risk of respiratory deterioration, along with the risk of buckling under a huge wave of COVID-19 infections and running out of medical capacity, may create a big challenge to HCP in Taiwan. The aims of this single-center study are to investigate the clinical manifestations and maternal–fetal outcomes in pregnant women with COVID-19 infection and to examine the effects of current obstetric management protocol on patient safety and on the prevention of nosocomial COVID-19 infection during peak outbreaks of the COVID-19 omicron variant in Taiwan in May 2022.

## 2. Materials and Methods

### 2.1. Data Collection and Definition

We retrospectively reviewed the medical records of confirmed and suspected COVID-19-positive mothers who delivered in NPIDRs and COVID-19 PCR-negative mothers who delivered in CDRs at Linkou CGMH between 1 May 2022 and 31 May 2022. Linkou CGMH, as the nearest and largest tertiary medical center to the Tao-Yuan, Hsinchu, and Miaoli cities in Taiwan, provides 24-hour open access through an emergency department system for referral of COVID-19 confirmed/suspected cases to seek negative pressure facilities or critical obstetric and neonatal care. The inclusion criterion is singleton pregnancy delivered at a gestational age of more than 34 weeks in our hospital during the study period. We excluded multiple gestations, obstetric deliveries at gestational age below 34 weeks, and major fetal anomalies from this study. All patients and their companions receive a government-funded 48-hour-validity COVID-19 PCR test before admission to facilitate patient triage.

We defined neonatal respiratory distress as newborns with the following symptoms: signs of tachypnea (rate >60 breaths per minute), cyanosis, expiratory grunting with chest retractions, and nasal flaring regardless of pulmonary or non-pulmonary causes [12]. The severity of neonatal respiratory distress is evaluated by Downes’ score [13,14]. Nosocomial COVID-19 infection among HCP is defined by an interval of 5 days between contact with COVID-19-positive confirmed cases and the first symptoms. In-hospital COVID-19 infection of patients and accompanying persons is defined as a delay of 5 days between admission and the first symptoms [15,16]. Each COVID-19 nosocomial event is stipulated to be uploaded to the hospital information system (HIS) for nosocomial infection surveillance.

All patient baseline characteristics, delivery information, maternal surgical outcomes (operative time, surgical blood loss, incidence of immediate postpartum hemorrhage, 14-day re-admission rate), and neonatal outcomes (1-minute Apgar score, 5-minute Apgar score, oxygen therapy requirement, antibiotic use, and the length of hospital stay) were extracted by medical record. Maternal surgical outcomes were compared separately based on delivery methods. The Institutional Review Board of Chang Gung Memorial Hospital approved this study (approval no. 202101275B0).

### 2.2. Obstetric Delivery Management Protocol

The obstetric delivery management protocol has been established since January 2020, conforming to the epidemic prevention guidance and recommendations of the T-CDC, and the protocol has been timely and practically modified in response to COVID-19 outbreak waves. To reduce the risk of in-hospital viral transmission and to preserve medical capacity, patient flow arrangements and the partition of hospital zones are applied based on the COVID-19 PCR test report, travel history, occupation history, contact history, and cluster information of individuals.

We defined a COVID-19 confirmed case as a positive result of a COVID-19 PCR test acquired from a nasopharyngeal swab within 14 days, regardless of clinical manifestation. The COVID-19 PCR test sample was collected from a nasopharyngeal swab by HCP, and the result was confirmed by an accredited lab. The cycle threshold (CT) value may indicate the viral load that an infected person harbors. The Department of Laboratory Medicine in our hospital states that if a positive signal is not seen after 39 cycles of amplification, then the PCR test is deemed negative. We defined a COVID-19 suspected case as an individual who has a negative COVID-19 PCR test but has contact history with COVID-19 confirmed cases in the last 7 days, has recent international travel history in the last 14 days, or has flu-like symptoms such as fever, shortness of breath, cough, sore throat, and loss of taste or smell.

Those identified as COVID-19 confirmed/suspected cases with the requirement of obstetric care are admitted to the NPIDRs. One obstetrician and one obstetrical nurse stand by at the nurse station of the isolated negative pressure delivery zone. The shift-based work is assigned independently from the CDRs to reduce the risk of in-hospital COVID-19 transmission. HCP put on PPE, including protective coveralls, protective gowns, and N95 respirators in the anteroom before entering the isolation room. Two obstetricians, four obstetrical nurses, and two neonatologists will enter the isolation zone to perform obstetric deliveries; one anesthesiologist and one assistant nurse will join the team if a cesarean section is indicated. Delayed cord clamping continues based on a lack of evidence in vertical transmission [3]. COVID-19-positive confirmed cases are transferred to the isolation ward (red zone) for postpartum care, while COVID-19 suspected cases are transferred to the step-down ward (yellow zone) after completing obstetric delivery. The postpartum care of COVID-19 confirmed/suspected cases follows the obstetric routine in our hospital. Breast milk feeding is encouraged, following appropriate breast and hand hygiene, and breast milk is collected and fed to the infant by designated caregivers. Patients who delivered through cesarean section will be discharged on the fourth postoperative day, and cases of vaginal delivery will be discharged on the third post-delivery day. The flowchart of obstetric management protocol is shown in Figure 1.

### 2.3. Neonatal Management Protocol

Two neonatologists put on PPE and enter the isolation zone of NPIDRs while performing obstetric delivery. The NPIDRs are equipped with infant warmers that are settled more than 2 m from the mother. If an open system ventilator support device such as nasal intermittent positive pressure ventilation, nasal continuous positive airway pressure (CPAP), or nasal cannula is indicated, the use of a high-efficiency particulate air (HEPA) filter between the mask and the T-piece resuscitator is required to minimize the risk of COVID-19 airborne transmission [3].

After initial management of the newborn, the baby is placed into an isolette inside the negative pressure delivery room. The newborn is then transferred to an isolated room in the neonatal intensive care unit (NICU) or intermediate care unit (IMU) through the isolette. All personnel in charge of standby, transfer, and care for the newborn wear aerosol precaution PPE. In terms of COVID-19 laboratory testing and isolation policy after birth, newborns of COVID-19-positive confirmed mothers will take COVID-19 PCR tests with the specimen obtained from the nasopharynx while admitted to the isolated unit; the subsequent test is arranged after 48 h from delivery. If two consecutive tests reveal negative results, the newborn is released from isolation. The newborn who has a positive result in the initial PCR test is transferred to the red zone ward, and a follow-up test is arranged repetitively in 48- to 72-h intervals until two consecutive negative results are obtained. On the other hand, newborns of COVID-19 suspected mothers will receive COVID-19 PCR tests once they are admitted to the isolated unit. If the PCR test is disclosed as negative, the newborn is released from quarantine. Video visits of neonates are provided for COVID-19 confirmed/suspected mothers to help relieve parental anxiety. Newborns with a negative result in the COVID-19 PCR test and no demands for medical care may be discharged. Although direct breastfeeding by COVID-19-infected mothers, immediate skin-to-skin contact after delivery, and rooming-in were not associated with an increased risk of newborn COVID-19 test positivity based on current evidence [4,17,18], the above hospital neonatal care practices are not allowed in current perinatal practice in our hospital. The flowchart of neonatal management protocol is demonstrated in Figure 1.

### 2.4. Statistics

Statistical analyses were performed with the software package MedCalc (MedCalc Software Ltd., Ostend, Belgium) and IBM SPSS (SPSS Inc., Chicago, IL, USA). The normality of the data was assessed with the Shapiro–Wilk test. Student’s *t*-test or the Mann–Whitney U-test was used to compare the continuous variables between groups. Qualitative data were compared using the χ^2^ test or Fisher’s exact test. A *p*-value of less than 0.05 was considered statistically significant.

## 3. Results

### 3.1. Patient Characteristics

A total of 213 obstetric deliveries of singleton pregnancy with gestational age ≥34 weeks are included in this study. The participant selection workflow is shown in Figure 2. A total of 45 deliveries were performed in NPIDRs due to a confirmed positive COVID-19 PCR test (*n* = 41) or suspected COVID-19 positive status (*n* = 4). Of those, 18 cases were cesarean sections and 25 cases were vaginal deliveries; 1 case was an operative vaginal delivery, and 1 case was a vaginal birth after cesarean (VBAC). One hundred and sixty-eight deliveries with negative COVID-19 PCR tests were performed in the CDRs; of those, 64 cases were cesarean deliveries, 93 cases were vaginal deliveries, 10 cases were operative vaginal deliveries, and 1 case was VBAC. The patient baseline demographics and delivery information are summarized in Table 1. Detailed information on maternal medical diseases in the two groups is listed in Appendix A. Distribution regarding delivery methods showed no significant difference between groups. There was no statistical difference in patient baseline characteristics (age, body mass index (BMI), gestational age at enrollment, maternal medical diseases, gestational diabetes mellitus (GDM), and gestational hypertension disease) between groups. Higher composition of primigravida was noted in the CDR group (58.0% vs. 37.8%, *p* = 0.016, respectively). Thirty-one cases (68.9%) of COVID-19 confirmed/suspected cases were referred from community hospitals. Thirty-two (78.0%) COVID-19-positive confirmed cases presented with mild upper-airway symptoms; cough (43.9%), sore throat (39.0%), and fever (22.0%) were the predominant symptoms in the present study. Nine (22.0%) COVID-19-positive confirmed mothers were asymptomatic. None of the COVID-19-infected mothers developed severe acute respiratory syndrome, and none of them required ventilator support, administration of anti-viral medication, or intensive care unit admission. Patient information and the clinical manifestation of deliveries performed in the NIPDRs are summarized in Table 2.

### 3.2. Maternal Surgical Outcomes

Maternal surgical outcomes were compared separately after pairing with delivery modes: vaginal delivery and cesarean section with the same obstetric indications.

In subgroup analyses of vaginal delivery, patient characteristics, including age, BMI, gestational age upon admission, proportion of primigravida, maternal medical diseases, GDM, and gestational hypertension disease, were similar between deliveries in the NPIDR and CDR groups. To maintain the medical capacity of the negative pressure facilities, only COVID-19-positive confirmed/suspected patients with emergent obstetric demands such as labor signs, unstable maternal conditions, or fetal distress were admitted to the NPIDRs. In terms of indication for admission, a higher percentage of labor signs (66.7% vs. 25.0%, *p* < 0.001, respectively) with lower proportions of planned labor induction (14.8% vs. 63.5%, *p* < 0.001, respectively) were observed in the NPIDR group compared to the CDR group.

Information regarding maternal surgical outcomes of vaginal delivery is summarized in Table 3. Ninety-four patients (90.4%) who delivered in the CDRs were provided painless labor through epidural analgesia; meanwhile, none of the deliveries in the NPIDRs received a painless labor service. Both total length of labor (300.5 min vs. 190.0 min, *p* = 0.003, respectively) and duration of stay in the delivery units (1119.5 min vs. 547.0 min, *p* = 0.001, respectively) were significantly shorter in the NPIDR group. Median surgical blood loss, median operative time, and rate of third- and fourth-degree perineal laceration after vaginal delivery showed no significant difference between the two groups. One mother was admitted to our ward again within 14 days from discharge due to delayed postpartum hemorrhage after vaginal delivery in the CDR group, while one re-admission within 14-day event occurred in a COVID-19-infected mother due to postpartum endometritis after vaginal delivery. The 14-day re-admission rate was not significantly different between the two groups of mothers.

To compare maternal surgical outcomes of cesarean section, patients were divided into groups based on different indications of cesarean section; thus, cesarean sections due to obstructive labor, placenta previa, chorioamnionitis, and maternal request were excluded from subgroup comparative analyses (Figure 2). A total of 18 and 51 cases of cesarean delivery were enrolled in subgroup comparative analyses in the NPIDR and CDR groups, respectively. Patient characteristics regarding age, BMI, gestational age upon admission, composition of primigravida, maternal medical disease, GDM, and gestational hypertension disease were similar between subgroups. (Table 4) The major indication of cesarean section in both groups was due to previous uterine surgery; moreover, the distribution of each cesarean delivery indication showed no significant difference between the groups. The total cesarean section rate was not significantly different between the two groups of patients. Surgical blood loss, operative time, and incidence of immediate postpartum hemorrhage disclosed nonsignificant differences regardless of indications for cesarean section. Subgroup analyses of maternal outcomes of cesarean delivery are shown in Table 4. Similar trends of indifferent maternal surgical outcomes could be found while pairing each case with the same indications of cesarean section (Appendix A). 

### 3.3. Perinatal Outcomes

Perinatal outcomes and clinical presentations of neonates are presented in Table 5. Median neonatal birth weight showed no significant difference between obstetric deliveries performed in NPIDRs and CDRs. All babies delivered in NPIDRs presented with an Apgar score of >7 at 1 min, and three (1.8%) neonates were born with an Apgar score at 1 min of <7 in the CDR group (*p* = 1.000). After initial management of the newborns, all babies in our study had Apgar scores at 5 min of more than 7. Higher rate of meconium-stained amniotic fluid (MSAF) was found in deliveries performed in NPIDRs than in CDRs (7 (4.2%) vs. 10 (22.2%), *p* < 0.001, respectively). Of the 11 babies from NIPDRs with MSAF were all born by COVID-19-positive confirmed mothers, 2 babies were delivered by emergent cesarean section due to non-reassuring fetal heart rate tracing and 9 babies were delivered by vaginal delivery; 3 neonates had meconium aspiration syndrome (MAS). More newborns delivered in NPIDRs had neonatal respiratory distress comparing to those born in CDRs (18 (40.0%) vs. 18 (10.7%), *p* < 0.001, respectively). Eighteen newborns who were delivered in NPIDRs with neonatal respiratory distress were all born by COVID-19-positive confirmed mothers; eleven were born by vaginal delivery and seven by cesarean section. Of all babies who presented with neonatal respiratory distress, those who were delivered in NPIDRs had a higher rate of oxygen therapy requirement than those delivered in CDRs (12 (26.7%) vs. 16 (9.5%), *p* = 0.003, respectively). Downes’ score was applied to evaluate the severity of neonatal respiratory distress; the median Downes’ score showed no significant difference between groups (*p* = 0.209), and most neonates who had respiratory distress was classified as mild-to-moderate respiratory distress, with Downes’ scores ranging from 2 to 5. Detailed information on the final diagnosis in newborns with neonatal respiratory distress and their demands for oxygen therapy are shown in Appendix A. Higher risk of newborn intensive care unit admission (14/168 (8.3%) vs. 25/45 (55.6%), *p* < 0.001) and longer median neonatal hospitalization (4 vs. 5 days, *p* ≤ 0.001) were also observed from deliveries in the NPIDR group than the CDR group.

Based on sequential reports of postnatal COVID-19 PCR tests, no vertical transmission was found.

### 3.4. Efforts on Control of COVID-19 In-Hospital Transmission

Looking at the records of nosocomial infection surveillance in the HIS, no nosocomial COVID-19 infection was reported among patients and HCP in charge of peripartum care for COVID-19-positive confirmed/suspected cases and their newborns during the study period.

## 4. Discussion

Our study demonstrates a poorer short-term neonatal outcome regarding higher NICU admission rate, increased incidence of MSAF, and higher risk of mild neonatal respiratory distress in cases delivered in NPIDRs than in CDRs under the current practice of obstetric delivery management protocol for the COVID-19 omicron variant pandemic. However, the maternal surgical outcomes showed no significant difference between the two groups of patients. Previous reports have found that pregnancy increases the susceptibility and severity of COVID-19 infection, and pregnant women with COVID-19 infection are more likely to be admitted to the intensive care unit or require invasive ventilation, accompanied by increasing maternal death compared to non-pregnant women of reproductive age with COVID-19 infection [18,19,20]. However, the omicron variant tends to be less aggressive in causing severe acute respiratory syndrome [21,22,23]. According to the T-CDC report, 99.53% of infected cases were either asymptomatic or with mild symptoms in the omicron variant epidemic stage in Taiwan from 1 January to 5 August 2022. Our report also found that 78% of cases had mild symptoms and 22% of cases were asymptomatic, and there was no severe form in pregnant women infected with the COVID-19 omicron variant during the study period. Different pre- and post-exposure risk factors may affect the outcomes of COVID-19 infection in pregnancy, including pre-existing maternal comorbidities, underlying chronic hypertension, pre-pregnancy diabetes, advanced maternal age, high BMI, accessibility to healthcare during outbreaks, and different COVID-19 coronavirus variants exposure. The baseline characteristics of pregnant women who delivered in NPIDRs and in CDRs were similar; hence, a COVID-19 omicron variant infection does not worsen maternal outcomes.

Respiratory distress is defined as any signs of increased work in the breathing of neonates, affecting 5% to 7% of all term newborns [24,25,26]. Neonatal respiratory distress is the most common reason that a newborn is admitted to the NICU. Risk factors for the occurrence of neonatal respiratory distress include preterm delivery, MSAF, cesarean section delivery, maternal diabetes, chorioamnionitis, oligohydramnios, or structural lung anomalies [12], and the incidence is increased even in modest prematurity [24,25,26]. In this study, we excluded cases of delivery at gestational age below 34 weeks, and we found the incidence of neonatal respiratory distress and neonatal NICU admission was higher in deliveries in NPIDRs than in CDRs. Increased stress in pregnant women is accompanied by higher risks of neonatal respiratory distress [27]. In the current policy, pregnant women delivered in NPIDRs were not allowed to have a companion or visitors and suffered from infant separation, but deliveries in CDRs were under no restrictions. The isolated nature of the NPIDRs may increase the psychological stress levels of pregnant women, while prenatal maternal stress might increase the risk of poorer perinatal outcomes [28,29,30,31,32]. Another stress source may be maternal COVID-19 infection [33,34]. We assumed that pregnant women who delivered in the NPIDRs in our series may have had more stress than pregnant mothers who delivered in the CDRs due to the isolated nature of NPIDRs and the maternal infection of COVID-19, thus resulting in higher rates of neonatal respiratory distress and MSAF, accompanied by higher NICU admission rates.

Increased cesarean delivery rates in COVID-19-positive confirmed mothers have been observed in previous studies [17,18,35,36]. The major contributing factor for the elevated cesarean section rate in COVID-19-positive confirmed cases may be maternal symptomatic COVID-19 infection with severe features, accompanied by medically indicated preterm delivery [7,17,35,36,37,38]. None of the pregnant women in our series, who had severe forms of COVID-19 infection, could explain why the total cesarean section rate was not higher in the COVID-19 confirmed/suspected group in the present study.

The use of PPE during the operation is time-consuming and makes exposure to the operative field more difficult; thus, it probably has adverse effects on the operative time and surgical outcomes [39]. In our study, we investigated the impact of PPE use on maternal surgical outcomes after pairing it with different delivery modes. Surgical blood loss, operative time, and incidence of immediate postpartum hemorrhage showed no significant differences between groups in cesarean delivery and in vaginal birth. The time of transfer to the operating room is short in our daily practice because the NPIDRs in our hospital are comprised of both vaginal delivery rooms and operative rooms. A regular comprehensive personnel training program for obstetricians, neonatologists, and anesthesiologists is essential to achieve rapid reaction and in-time management for the delivery of high-risk pregnancies.

We performed COVID-19 PCR tests by nasopharyngeal swab for neonates 1 to 2 times within 48 h from delivery to examine vertical transmission and to preclude the possibility of a positive result due to postpartum horizontal transmission. No vertical transmission was observed based on the current protocol of specimen collection and PCR testing in our study. To precisely identify utero-placental transmission of COVID-19 infection, various fetal tissue specimen collection methods and serologic tests have been investigated in previous studies, with different rates of positive results [40]. Although the heterogeneous timing of neonatal COVID-19 PCR tests from delivery has been described in previous studies, the pooled proportion of COVID-19 PCR positivity in neonates by nasopharyngeal swab only ranged from 2.0% to 4.4% [7,17,40,41,42]. The small sample size of patients in our study could explain why no vertical transmission was found.

Our study had some limitations. First, this is a retrospective and single-institution study, so the number of cases is limited. Second, we only have the neonatal PCR test from nasopharyngeal swabs to evaluate vertical transmission, and specimen collection of other fetal tissues, including placental tissue, cord-blood, fetal rectal swabs, amniotic fluid, or other neonatal serologic tests, might provide more information regarding the utero-placental transmission of COVID-19. Despite the above limitations, our study shares the largest case series of COVID-19-positive confirmed mothers and their newborns in Taiwan during the COVID-19 omicron variant pandemic period. Furthermore, our study demonstrates the current strategy in our hospital, including COVID-19 PCR screening before admission, patient partition, TOCC risk assessment, and the establishment of practice protocol to ensure a low rate of associated maternal–fetal comorbidities, provide good accessibility to healthcare services in neighborhood areas, reduce the risk of hospital-based COVID-19 transmission, and maintain medical capacity.

## 5. Conclusions

Our study shows that obstetric deliveries for COVID-19 omicron variant positive and suspected cases performed in NPIDRs with the rational use of PPE are not associated with adverse maternal surgical outcomes, regardless of delivery methods. Short-term perinatal outcomes were poorer in neonates of COVID-19 omicron variant positive confirmed and suspected mothers who delivered in NPIDRs compared to COVID-19-negative mothers who delivered in CDRs, with higher rates of neonatal respiratory distress, MSAF, and NICU admission. These findings might be due to more perinatal stress in obstetric deliveries performed in NPIDRs. Reasonable use of PPE in NPIDRs could effectively prevent COVID-19 nosocomial infection during the delivery of COVID-19 omicron-variant-infected pregnant women.

## Figures and Tables

**Figure 1 jcm-11-05441-f001:**
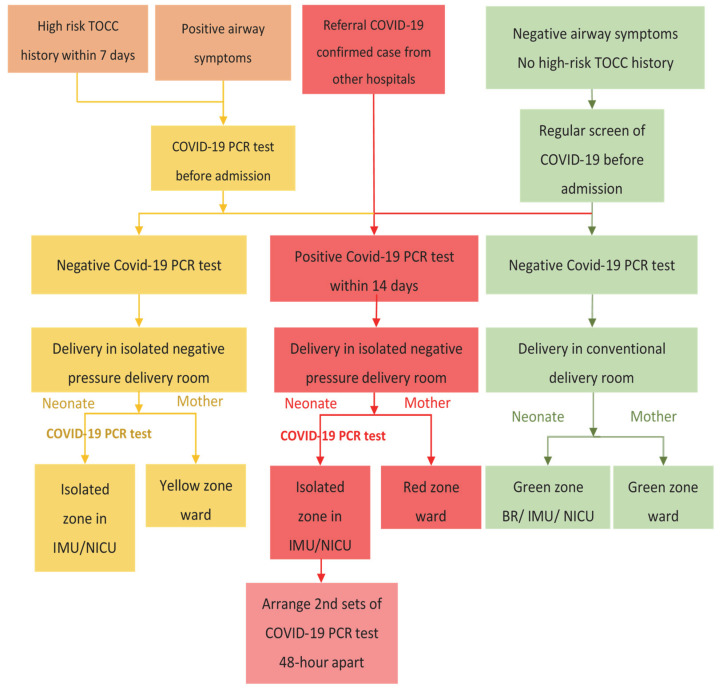
Flow chart of management of COVID-19 confirmed or suspected mothers and neonates. BR, baby room; COVID-19, coronavirus disease 2019; IMU, intermediate care unit; NICU, neonatal intensive care unit; TOCC, travel, occupational contact, and cluster.

**Figure 2 jcm-11-05441-f002:**
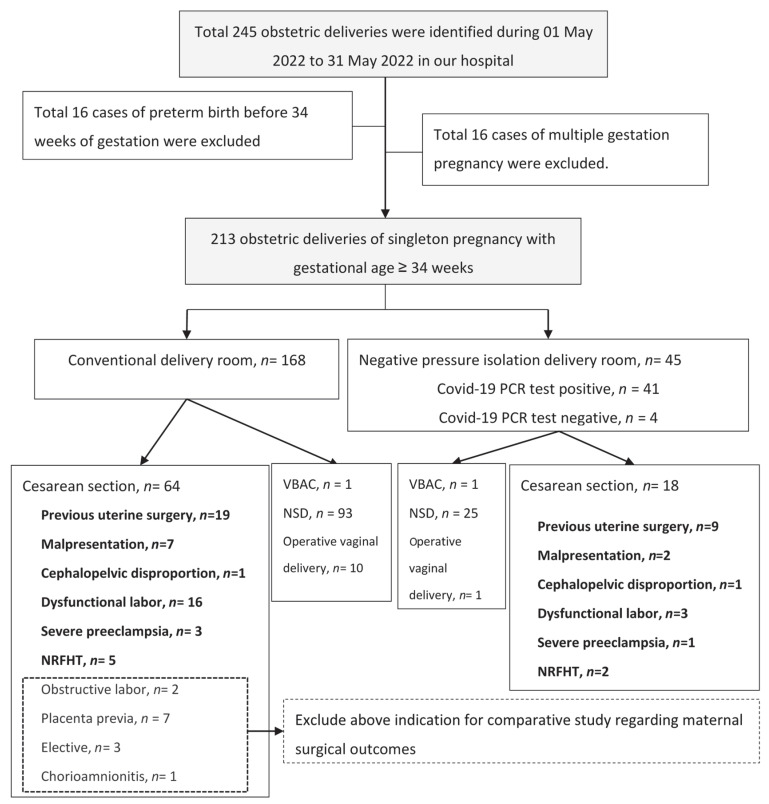
Flowchart describing the study population selection. NSD, natural spontaneous delivery; NRFHT, non-reassuring fetal heartbeat tracing; PCR, polymerase chain reaction; VBAC, vaginal birth after cesarean.

**Table 1 jcm-11-05441-t001:** Patient baseline characteristics and delivery information of study population.

	Delivery Room (*n* = 168)		Isolated Negative Pressure Room (*n* = 45)		*p*-Value
Age (median)	32 (19–44)		30 (18–44)		0.082 ^†^
BMI(median)	26.6 (17.9–47.6)		27.5 (19.6–43.6)		0.218 ^†^
Gravida (median)	2 (1–6)		2 (1–8)		0.022 ^†^
Primi-gravida	98 (58.3%)		17 (37.8%)		0.014 *
Gestational age by weeks (median)	38 3/7 (34 3/7–40 6/7)		38 4/7 (34 4/7–40 2/7)		0.305 ^†^
GDM	19 (11.3%)		4 (8.9%)		0.791 ^◊^
PIH	6 (3.6%)		4 (8.9%)		0.224 ^◊^
Preeclampsia	7 (4.2%)		2 (4.4%)		1.000 ^◊^
IUGR	3 (1.8%)		0		1.000 ^◊^
Medical diseases	21 (12.5%)		2 (4.4%)		0.176 ^◊^
COVID-19 PCR test results	Negative	168	Negative	4	-
	Positive	0	Positive	41	-
Delivery Mode					
Vaginal delivery	93 (55.4%)		25 (55.6%)		0.981 *
Operative vaginal delivery	10 (6.0%)		1 (2.2%)		0.464 ^◊^
VBAC	1 (0.6%)		1 (2.2%)		0.379 ^◊^
Cesarean section	64 (38.1%)		18 (40.0%)		0.816 *
Primary CS	38 (22.6%)		7 (15.6%)		0.304 *
Emergent CS	16 (9.5%)		9 (20.0%)		0.053 *
Total CS rate	38.1%		40.0%		0.816 *
Primary CS rate	38/(38 + 93 + 10) = 26.8%		7/(25 + 1 + 7) = 21.2%		0.312 *

BMI, body mass index; COVID-19, coronavirus disease 2019; CS, cesarean section; GDM, gestational diabetes mellitus; IUGR, intrauterine growth restriction; PCR, polymerase chain reaction; PIH, pregnancy-induced hypertension; VBAC, vaginal birth after cesarean. ^†^ Mann–Whitney U-test; * chi-square test; ^◊^ Fisher’s exact test.

**Table 2 jcm-11-05441-t002:** Clinical characteristics of patients delivered in negative pressure isolation delivery rooms.

	COVID-19 Confirmed Case, *n* = 41	Close Contact with Confirmed Case, *n* = 4
COVID-19 PCR test	Positive = 41	Positive = 0
Referral cases	29	3
Symptoms of COVID-19 infection		
Fever	9	0
Cough	18	0
Sore throat	16	0
Muscle or body aches	4	0
Congestion or rhinorrhea	3	0
Shortness of breath	0	0
Asymptomatic	9	4

COVID-19, coronavirus disease 2019; PCR, polymerase chain reaction.

**Table 3 jcm-11-05441-t003:** Patient characteristics and maternal surgical outcomes in the vaginal delivery group (*n* = 131).

	Delivery Room, *n* = 104	Isolated Negative Pressure Room, *n* = 27	*p*-Value
Age (median)	31.5 (19–43)	30 (20–41)	0.109 ^†^
BMI (median)	25.50 (17.9–47.6)	27.34 (19.6–35.5)	0.276 ^†^
Gravida (median)	2 (1–6)	2 (1–8)	0.021 ^†^
Primigravida	60 (57.7%)	10 (37.0%)	0.056 *
Gestational age by weeks (median)	38 4/7 (34 1/7–40 6/7)	38 5/7 (34 3/7–40 2/7)	0.393 ^†^
GBS colonization	23 (22.1%)	6 (22.2%)	0.839 ^◊^
GDM	11 (10.6%)	2 (7.4%)	1.000 ^◊^
PIH	2 (1.9%)	1 (3.7%)	0.503 ^◊^
Preeclampsia	4 (3.7%)	0 (0.0%)	0.583 ^◊^
IUGR	3 (3.9%)	0 (0.0%)	1.000 ^◊^
Medical diseases	10 (9.6%)	1 (3.7%)	0.459 ^◊^
Delivery Mode			
NSD	93 (89.4%)	25 (92.6%)	0.625 *
Operative vaginal delivery	10 (9.6%)	1 (3.7%)	0.459 ^◊^
VBAC	1 (1.00%)	1 (3.7%)	0.371 ^◊^
Trial of Labor			
Rupture of membrane	12 (11.5%)	5 (18.5%)	0.409 ^◊^
In labor	26 (25.0%)	18 (66.7%)	<0.001 *
Induction of labor	66 (63.5%)	4 (14.8%)	<0.001 ^◊^
Maternal Surgical Outcome			
Change to CS (%)	16 (15.4%)	0 (0.0%)	<0.001 ^†^
1st stage of labo(min)	242.5 (15–1050)	160 (30–930)	0.005 ^†^
2nd stage of labor (min)	48 (6–217)	27 (4–108)	0.009 ^†^
3rd stage of labor (min)	1 (1–13)	1 (1–6)	0.416 ^†^
Total labor (min)	300.5 (35–1077)	190 (35–978)	0.003 ^†^
Time of stay in delivery unit (min)	1119.5 (148–3967)	547 (133–1843)	0.001 ^†^
Epidural analgesia (%)	94 (90.4%)	0 (0.0%)	<0.001 ^◊^
Blood loss (mL)	200 (100–400)	200 (200–850)	0.102 ^†^
Operative time (min)	24 (12–54)	25 (19–54)	0.080 ^†^
3rd- to 4th-degree lacerations (%)	15 (14.4%)	2 (7.4%)	0.522 ^◊^
Re-admission in 14 days	1 (1.0%)	1 (3.7%)	0.369 ^◊^

BMI, body mass index; CS, cesarean section; GBS, group B streptococcus; GDM, gestational diabetes mellitus; IUGR, intrauterine growth restriction; NSD, natural spontaneous delivery; PIH, pregnancy-induced hypertension; VBAC, vaginal birth after cesarean. ^†^ Mann–Whitney U-test; * chi-square test; ^◊^ Fisher’s exact test.

**Table 4 jcm-11-05441-t004:** Patient characteristics and maternal surgical outcomes in paired cesarean section group (*n* = 69).

	Delivery Room, *n* = 51	Isolated Negative Pressure Room, *n* = 18	*p*-Value
Age (median)	34 (23–44)	32 (18–44)	0.377 ^†^
BMI (median)	27.5 (21.3–38.1)	28.4 (22.2–43.6)	0.280 ^†^
Gravida (median)	2 (1–5)	2 (1–6)	0.348 ^†^
Para (median)	1 (1–4)	2 (1–4)	0.358 ^†^
Abortus (median)	0 (0–4)	0 (0–4)	0.694 ^†^
Gestational age by weeks (median)	38 1/7 (34 2/7–40 1/7)	38 1.5/7 (35 3/7–40)	0.722 ^†^
GDM	6 (11.8%)	2 (11.1%)	1.000 ^◊^
PIH	2 (3.9%)	3 (16.7%)	0.107 ^◊^
Preeclampsia	2 (3.9%)	2 (11.1%)	0.277 ^◊^
IUGR	0	0	-
Medical diseases	6 (11.8%)	1 (5.6%)	0.665 ^◊^
Emergent cesarean section	15 (29.4%)	9 (50.0%)	0.118
Cesarean Section Indications			
Previous uterine surgery	19 (37.3%)	9 (50.0%)	0.347 *
Malpresentation	7 (13.7%)	2 (5.6%)	1.000 ^◊^
Cephalopelvic disproportion	1 (2.0%)	1 (5.6%)	0.457 ^◊^
Dysfunctional labor	16 (31.4%)	3 (11.1%)	0.358 ^◊^
Severe preeclampsia	3 (5.9%)	1 (5.6%)	1.000 ^◊^
NRFHT	5 (9.8%)	2 (11.1%)	1.000 ^◊^
Maternal Outcome Paired with Indications of Cesarean Deliveries			
Blood loss (mL)	500 (300–1800)	425 (200–1100)	0.365 ^†^
Postpartum hemorrhage (%)	6 (11.8%)	2 (11.1%)	1.000 ^◊^
Operative time (min)	81 (39–126)	91 (64–127)	0.111 ^†^

BMI, body mass index; GDM, gestational diabetes mellitus; IUGR, intrauterine growth restriction; NRFHT, non-reassuring fetal heartbeat tracing; PIH, pregnancy-induced hypertension. ^†^ Mann–Whitney U-test; * chi-square test; ^◊^ Fisher’s exact test.

**Table 5 jcm-11-05441-t005:** Perinatal outcome and clinical presentation of neonates in our study population.

	Delivery Room, *n* = 168	Isolated Negative Pressure Room, *n* = 45	*p*-Value
Birth weight (gm)	3040 (1765–3825)	3090 (2006–4010)	0.215 ^†^
MSAF	7 (4.2%)	10 (22.2%)	<0.001 *
Apgar score’1 < 7	3 (1.8%)	0	1.000 ^◊^
Apgar score’5 < 7	0	0	-
NICU admission	14 (8.3%)	25 (55.6%)	<0.001 *
Neonatal respiratory distress	18 (10.7%)	18 (40.0%)	<0.001 *
Oxygen therapy	16 (9.5%)	12 (26.7%)	0.003 *
Median Downes’ score	4 (2–5)	3 (2–5)	0.209 ^†^
Downes’ score < 4	5 (27.8%)	9 (52.9%)	0.176 ^◊^
Downes’ score 4–7	13 (72.2%)	8 (47.2%)	0.134 *
Downes’ score > 7	0	0	-
Neonatal fever	10 (6.0%)	1 (2.2%)	0.464 ^◊^
Antibiotic use	26 (15.5%)	17 (37.8%)	0.001 *
Median hospitalization (days)	4 (3–19)	5 (3–12)	<0.001 ^†^

MSAF, meconium-stained amniotic fluid; NICU, neonatal intensive care unit. ^†^ Mann–Whitney U-test; * chi-square test; ^◊^ Fisher’s exact test.

## Data Availability

The datasets obtained and analyzed in this study are available from the corresponding author upon reasonable request.

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
