# Peer review of "Maternal–Neonatal Outcomes of Obstetric Deliveries Performed in Negative Pressure Isolation Rooms during the COVID-19 Omicron Variant Pandemic in Taiwan: A Retrospective Cohort Study of a Single Institution"

_jcm, 2022, doi:10.3390/jcm11185441_

Round 1

Reviewer 1 Report

It is a very clear case-control study about maternal and neonatal outcomes of obstetric deliveries performed in labor ward rooms and negative pressure isolation rooms during the Covid-19 Omicron 3 variant pandemic in Taiwan. Very complete data are presented .  

Minor spelling errors must be corrected, for eg. line 77 previous studies instead of previous study. 

One question from table 4 , operative time for C- section was 81 and respectively 91 minutes. 

Author Response

It is a very clear case-control study about maternal and neonatal outcomes of obstetric deliveries performed in labor ward rooms and negative pressure isolation rooms during the Covid-19 Omicron 3 variant pandemic in Taiwan. Very complete data are presented . 

  1. Minor spelling errors must be corrected, for eg. line 77 previous studies instead of previous study.

Answer:We are very grateful to your comments for the manuscript. According with your advice, we will correct the spelling errors to improve the quality of our manuscript.

  1. One question from table 4, operative time for C- section was 81 and respectively 91 minutes.

Answer:Thank you for the time spent reviewing our manuscript. Different indications of cesarean section carry various degrees of difficulty, and might have impact on the operative time. Therefore, we paired patients with same indications for cesarean delivery in our study population, and found the median operative time of cesarean section in isolated negative pressure delivery room and in conventional delivery room was 91 minutes and 81 minutes, respectively. Although the discrepancy of operative times between two groups showed no statistical significance in our study, longer operative time for cesarean section performed in isolated negative pressure delivery room was consumed.  

Reviewer 2 Report

The authors of “Maternal-neonatal outcomes of obstetric deliveries performed in negative pressure isolation rooms during Covid-19 Omicron variant pandemic in Taiwan: a retrospective cohort study in a single institution” performed a retrospective study of the medical records of pregnant COVID-19 positive confirmed, and COVID-19 suspected cases delivered in NPIDR, and COVID-19 PCR negative mothers delivered in CDR at Linkou CGMH during 1 May 2022 to 31 May 2022.

There are 213 obstetric deliveries of singleton pregnancy with gestational age ≥ 34 weeks. A total of 45 deliveries were performed in NPIDR due to confirmed positive COVID-19 PCR test (n= 41), or suspected COVID-19 positive status (n=4); of 18 cases were cesarean sections, and 25 cases were vaginal delivery, 1 was an operative vaginal delivery, and 1 was VBAC. 168 deliveries with negative COVID-19 PCR test were performed in CDR; 64 cases were cesarean deliveries, 93 cases were vaginal delivery, 10 were operative vaginal deliveries, and 1 was vaginal birth after cesarean (VBAC).

31 cases (68.89%) of COVID-19 confirmed/suspected cases were referred from community hospitals. 32(78.0%) COVID-19 positive confirmed cases presented with mild upper-airway symptoms; cough (43.9%), sore throat (39.0%), and fever (22.0%). 9 (22.0 %) COVID-19 positive confirmed mothers were asymptomatic. 

40% of newborns delivered in NPIDR from mothers with confirmed positive COVID-19 had neonatal respiratory distress compared to those in 10.71% CDR. 

They concluded that pregnant women delivered in NPIDR might be under more stress than pregnant mothers delivered in CDR due to the isolated nature of NPIDR and maternal infection of COVID-19, thus resulting in a higher rate of neonate respiratory distress and MSAF than complicated with higher NICU admission rate.

Many other studies have observed the increased cesarean delivery rate in COVID-19-positive confirmed mothers.

I congratulate the authors for their work. This study shared the most prominent case series of COVID-19 positive confirmed mother and their newborns, demonstrated the current strategy in the hospital for COVID-19 PCR screening before admission, patient partition, TOCC risk assessment, and the establishment of practice protocol ensured a low rate of associated maternal-fetal comorbidities, good accessibility to healthcare services in the neighborhood area, and reduce the risk of hospital-based COVID-19 transmission to maintain medical capacity.

 Some misspellings of English. Please review the manuscript with an English native.

Author Response

Reviewer 2

The authors of “Maternal-neonatal outcomes of obstetric deliveries performed in negative pressure isolation rooms during Covid-19 Omicron variant pandemic in Taiwan: a retrospective cohort study in a single institution” performed a retrospective study of the medical records of pregnant COVID-19 positive confirmed, and COVID-19 suspected cases delivered in NPIDR, and COVID-19 PCR negative mothers delivered in CDR at Linkou CGMH during 1 May 2022 to 31 May 2022.

There are 213 obstetric deliveries of singleton pregnancy with gestational age ≥ 34 weeks. A total of 45 deliveries were performed in NPIDR due to confirmed positive COVID-19 PCR test (n= 41), or suspected COVID-19 positive status (n=4); of 18 cases were cesarean sections, and 25 cases were vaginal delivery, 1 was an operative vaginal delivery, and 1 was VBAC. 168 deliveries with negative COVID-19 PCR test were performed in CDR; 64 cases were cesarean deliveries, 93 cases were vaginal delivery, 10 were operative vaginal deliveries, and 1 was vaginal birth after cesarean (VBAC).

31 cases (68.89%) of COVID-19 confirmed/suspected cases were referred from community hospitals. 32(78.0%) COVID-19 positive confirmed cases presented with mild upper-airway symptoms; cough (43.9%), sore throat (39.0%), and fever (22.0%). 9 (22.0 %) COVID-19 positive confirmed mothers were asymptomatic.

40% of newborns delivered in NPIDR from mothers with confirmed positive COVID-19 had neonatal respiratory distress compared to those in 10.71% CDR.

They concluded that pregnant women delivered in NPIDR might be under more stress than pregnant mothers delivered in CDR due to the isolated nature of NPIDR and maternal infection of COVID-19, thus resulting in a higher rate of neonate respiratory distress and MSAF than complicated with higher NICU admission rate.

Many other studies have observed the increased cesarean delivery rate in COVID-19-positive confirmed mothers.

I congratulate the authors for their work. This study shared the most prominent case series of COVID-19 positive confirmed mother and their newborns, demonstrated the current strategy in the hospital for COVID-19 PCR screening before admission, patient partition, TOCC risk assessment, and the establishment of practice protocol ensured a low rate of associated maternal-fetal comorbidities, good accessibility to healthcare services in the neighborhood area, and reduce the risk of hospital-based COVID-19 transmission to maintain medical capacity.

 Some misspellings of English. Please review the manuscript with an English native.

Answer:We acknowledge the reviewer’s comments and suggestions very much, which are valuable in improving the quality of our manuscript. We will review the manuscript with an English native.